# RealDPO: Real or Not Real, that is the Preference

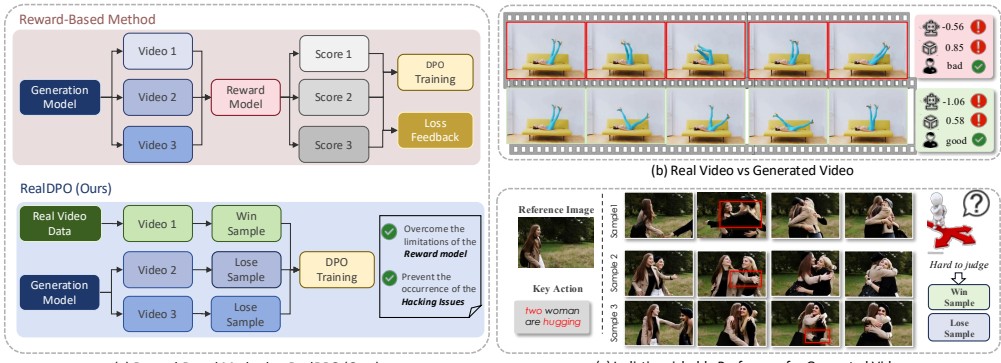

Figure 1: ***Can we align video generative models using real data as preference data without a reward model?*** (a) Comparison between using the reward model to score synthetic data for preference learning and our RealDPO method, which uses high-quality real data as win samples. Our method avoids the limitations of the reward model and the associated hacking issues. (b) Comparison between the video generated by the pretrained model and the real video for the same scene. The three scores on the right represent the scores given by the reward model from VisionReward (Xu et al., 2024a), the human action metric from VBench (Huang et al., 2024a;b), and human preference, respectively. It can be observed that while the existing reward model and VBench can evaluate semantic correctness, they are limited in assessing human motion quality. (c) Three model-generated examples from the same prompt, each with different initial noise, exhibit poor limb interaction, making it challenging for human annotators to identify which sample should be chosen as the win sample for reward model training.

## Abstract

Video generative models have recently achieved notable advancements in synthesis quality. However, generating complex motions remains a critical challenge, as existing models often struggle to produce natural, smooth, and contextually consistent movements. This gap between generated and real-world motions limits their practical applicability. To address this issue, we introduce **RealDPO**, a novel alignment paradigm that leverages real-world data as positive samples for preference learning, enabling more accurate motion synthesis. Unlike traditional supervised fine-tuning (SFT), which offers limited corrective feedback, RealDPO employs Direct Preference Optimization (DPO) with a tailored loss function to enhance motion realism. By contrasting real-world videos with erroneous model outputs, RealDPO enables iterative self-correction, progressively refining motion quality. To support post-training in complex motion synthesis, we propose **RealAction-5K**, a curated dataset of high-quality videos capturing human daily activities with rich and precise motion details. Extensive experiments demonstrate that RealDPO significantly improves video quality, text alignment, and motion realism compared to state-of-the-art models and existing preference optimization techniques.

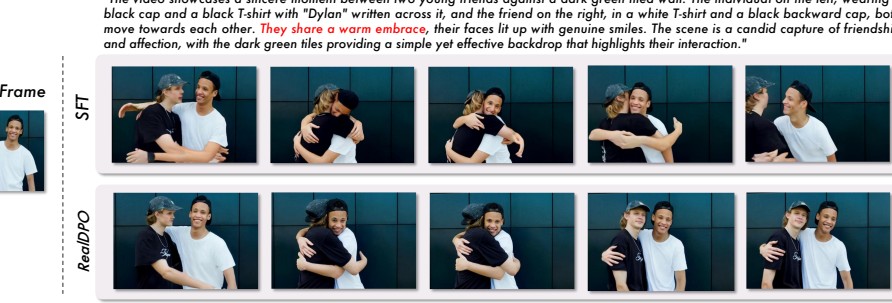

Figure 2: **RealDPO vs SFT.** A qualitative comparison between RealDPO and supervised fine-tuning (SFT). RealDPO demonstrates more natural motion generation. For more details regarding the comparison, please refer to the supplementary material.

# 1 INTRODUCTION

With the advancement in computational power and the availability of large-scale data, video generation models (Yang et al., 2024b; Guo et al., 2023; Blattmann et al., 2023; Li et al., 2024; Lin et al., 2024; Wang et al., 2024b; Xing et al., 2024; Zhang et al., 2023) have made significant progress, producing more realistic and diverse visual content. However, when it comes to generating complex motions, existing models still face considerable challenges in creating motion sequences that adhere to appear natural and smooth, and align with contextual consistency. This issue becomes especially prominent in the generation of human-centric daily activity motions. As shown in Fig. 1(b), even the results generated by the state-of-the-art DiT-based model CogVideoX-5B (Yang et al., 2024b) exhibit unnatural and unrealistic movements, failing to meet human preferences for natural, smooth, and contextually appropriate actions. This prompts us to further explore how to improve the realism and rationality of complex motion generation, particularly in the domain of human motion synthesis.

A straightforward solution is to collect a set of high-quality, real-world data specifically for supervised fine-tuning (SFT). However, relying exclusively on this dataset for SFT training presents certain limitations. During optimization, the model interprets the provided data as the sole correct reference, lacking awareness of where the original model's errors stem from. This narrow focus may result in overfitting and suboptimal performance in Fig. 2. A more effective strategy would be letting the model learn from its own mistakes. By utilizing the difference between real samples (positive data) and generative samples (negative data), we can explicitly highlight the model's errors and guide it to correct its behavior. This approach enables the model to progressively align its outputs with the desired actions represented by the positive samples, fostering continuous improvement through self-reflection. This idea aligns perfectly with Direct Preference Optimization (DPO) (Rafailov et al., 2023), a reinforcement learning technique used in training large language models, which leverages pair-wise win-lose data to guide the learning process.

In video generation, recent studies (Liu et al., 2024; Wang et al., 2024c; Xu et al., 2024a; Yuan et al., 2024; Zhang et al., 2024a) have explored training fine-grained reward models using human-annotated preference datasets, primarily through three ways: reward-weighted regression (RWR) (Wang et al., 2024c), direct preference optimization (DPO) (Liu et al., 2024), and gradient feedback (GF) (Yuan et al., 2024). However, these methods face some critical challenges when directly applied to action-centric video generation: (1) *Reward Hacking*: Video reward model is susceptible to reward hacking, where during the optimization process, human evaluations indicate a decline in video quality, yet the reward model continues to assign high scores. (2) *Scalability Issue*: Online approaches require decoding latent to pixel space, limiting their scalability for high-resolution video generation. (3) *Bias Propagation*: Multi-dimensional reward models may lose the ability to assess specific key metrics due to linear combinations of evaluation criteria. As shown in Fig. 1(b), the reward model cannot provide an accurate evaluation for complex motion. These limitations highlight the need for a more robust approach tailored to complex motion video generation, motivating our extension beyond traditional DPO frameworks.

To address these challenges, we propose **RealDPO**, a novel training pipeline for generating action-centric activity videos, as shown in Fig. 1(a). Unlike prior methods that rely on model-sampled pairwise comparisons, RealDPO leverages real-world video data as win samples, overcoming the ***Real Data Deficiency*** issue where only using synthetic data for preference learning fails to address the distribution errors inherent in the pre-trained generative model. More importantly, this approach significantly enhances the model's learning upper bound, enabling more accurate video generation. Without real video guidance, as shown in Fig. 1(c), all samples generated by pre-trained model exhibit poor limb interaction, making it hard for human annotators to identify the preferred win sample. Additionally, since RealDPO directly uses real data to guide the preference learning, it eliminates the need for an external reward function, thereby avoid reward hacking and bias propagation issues. Moreover, our naturally paired win-lose samples eliminate the need for decoding latent to pixel space during training, drastically reducing computational overhead. Inspired by Diffusion-DPO (Wallace et al., 2024), we design a tailored DPO loss specifically for the training objective of diffusion-based transformer architectures, enabling effective preference alignment. To support this training, we introduce **RealAction-5K**, a compact yet high-quality video dataset capturing diverse human daily actions. The dataset adheres to the principle of "less is more", emphasizing that RealDPO requires fewer high-quality samples in synergy with model-generated negative samples, whereas traditional supervised fine-tuning (SFT) methods typically requires more data to achieve better performance. Experiments demonstrate that RealDPO significantly improves video quality, text alignment, and action fidelity across diverse human action scenarios compared to pretrained baselines and other preference alignment methods. Our contributions are summarized as follows:

- We propose RealDPO, a novel training pipeline for action-centric video generation that leverages real-world data as preference signals to contrastively reveal and correct the model's inherent mistakes, addressing the limitations of existing reward models and preference alignment methods.

- We design a tailored DPO loss for our video generation training objective, enabling efficient and effective preference alignment without the scalability and bias issues of prior approaches.

- We introduce RealAction-5K, a compact yet high-quality curated dataset focused on human daily actions, specifically crafted to advance preference learning for video generation models and broader applications.

## 2 RELATED WORK

**Diffusion-Based Video Generation.** In recent years, diffusion-based video generation models have emerged continuously, primarily generating videos through user-provided text or image prompts. These models are broadly categorized into two architectures: U-Net and Diffusion Transformers (DiT). U-Net-based approaches (Blattmann et al., 2023; Wang et al., 2023; Chen et al., 2024; Guo et al., 2023) build upon the multi-stage down-sampling and up-sampling framework of image diffusion models, incorporating temporal attention layers to ensure temporal consistency. However, these methods face limitations in motion dynamics and content richness. Recently, Diffusion Transformer-based methods (Yang et al., 2024b; Li et al., 2024; Lin et al., 2024) have made significant improvements by combining 3D-VAE with diffusion transformers, using 3D full-attention layers to jointly learn spatial-temporal correlations, and enhance text encoders to handle complex prompts. These advancements have led to substantial improvements in fidelity, consistency, and scalability for longer video generation.

**Reinforce Learning in Image/Video Generation.** In large language models (LLMs), reward models are commonly used in Reinforcement Learning Human Feedback (RLHF), enabling LLMs to respond more naturally and generate more coherent text. Recently, there have been a series of studies(Xu et al., 2024b; Black et al., 2023; Wallace et al., 2024; Lee et al., 2023; Liang et al., 2024; Yang et al., 2024a; Clark et al., 2023; Fan et al., 2024) in image generation that incorporate human preferences into model evaluation and model alignment training, mainly focusing on improving the aesthetic quality of images. In video generation, the exploration so far is still quite limited. Most related works(Yuan et al., 2024; Wu et al., 2024; Wang et al., 2024c; Liu et al., 2024; Zhang et al., 2024a; Liu et al., 2025) are mainly focusing on using reward models trained on human-annotated synthetic data for preference learning on model-generated data. However, these methods have some limita-

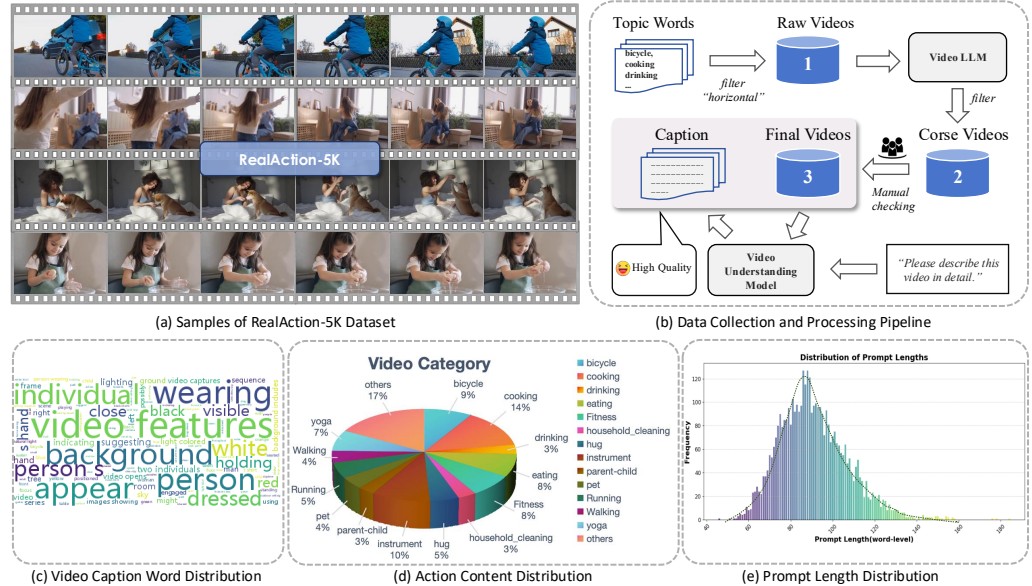

Figure 3: **Overview of the RealAction-5K Dataset.** (a) Samples of RealAction-5K Dataset (b) Data Collection and Processing Pipeline (c) Video Caption Word Distribution (d) Action Content Distribution (e) Prompt Length Distribution

tions. For example, training reward models may suffer from hacking issues, and multi-dimensional reward models might show reduced evaluation ability in specific domains. Additionally, relying entirely on synthetic data for preference learning could hinder the model's potential. Therefore, we propose a novel approach that transcends the limitations of reward models by incorporating real data for preference-aligned learning.

## 3  PRELIMINARIES

### 3.1  DENOISING PROCESS AS MULTI-STEP MDP

According to the definition in the Yang et al. (2024a), the denoising process in diffusion models can be formulated as a multi-step Markov Decision Process (MDP). Here, we provide a further explanation of state representations $\mathbf{s}_t$, probability transition $P$, and policy functions $\pi$, establishing a correspondence between video diffusion models and the MDP framework. This mapping enables a reinforcement learning perspective on the sampling process in video diffusion models. The detailed notation correspondence between the diffusion model and the MDP is as follows:

$$
\begin{aligned}
\mathbf{s}_t &\triangleq (\boldsymbol{c}, t, \boldsymbol{x}_t) \quad P(\mathbf{s}_{t+1} \mid \mathbf{s}_t, \mathbf{a}_t) \triangleq (\delta_{\boldsymbol{c}}, \delta_{t+1}, \delta_{\boldsymbol{x}_{t-1}}) \\
\mathbf{a}_t &\triangleq \boldsymbol{x}_{t-1} \quad\quad \pi(\mathbf{a}_t \mid \mathbf{s}_t) \triangleq p_\theta(\boldsymbol{x}_{t-1} \mid \boldsymbol{c}, t, \boldsymbol{x}_t) \\
\rho_0(\mathbf{s}_0) &\triangleq (p(\boldsymbol{c}), \delta_0, \mathcal{N}(\mathbf{0}, \mathbf{I})) \\
r(\mathbf{s}_t, \mathbf{a}_t) &\triangleq r((\boldsymbol{c}, t, \boldsymbol{x}_t), \boldsymbol{x}_{t-1})
\end{aligned}
\tag{1}
$$

where $\delta_x$ represents the Dirac delta distribution, and $t$ denotes the denoising timesteps. Leveraging this mapping, we can employ RL techniques to fine-tune diffusion models by maximizing returns. However, this approach requires a proficient reward model capable of adequately rewarding the noisy images. The task becomes exceptionally challenging, particularly when $t$ is large, and $\boldsymbol{x}_t$ closely resembles Gaussian noise, even for an experienced expert.

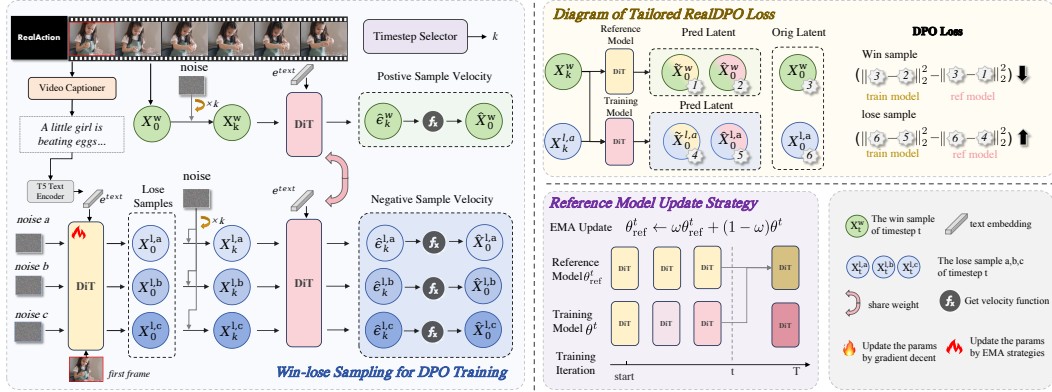

Figure 4: **The RealDPO Framework**. We use real data as the win samples in DPO, and illustrate the data pipeline on the left hand side. We present the RealDPO loss, and reference model update strategy on the right hand side.

## 3.2 DPO FOR DIFFUSION MDP

Direct Preference Optimization (DPO) (Rafailov et al., 2023) is a preference-based fine-tuning method that directly optimizes a model using human preference data, without requiring an explicit reward model. This approach is particularly advantageous as it avoids the complexities and potential biases introduced by learned reward models, making the optimization process more stable and interpretable. Given a dataset of preference-labeled pairs $\{(x, y_w, y_l)\}$, where $x$ is the input prompt, $y_w$ is the preferred (win) output, and $y_l$ is the less preferred (lose) output, DPO aims to maximize the likelihood ratio between the preferred and non-preferred samples while maintaining closeness to the pretrained model. The optimization objective can be formulated as:

$$L_{\text{DPO}}(\theta) = -\mathbb{E}_{c,x^w,x^l}\left[\log\sigma\left(\beta\log\frac{p_\theta(x^w|c)}{p_{\text{ref}}(x^w|c))} - \beta\log\frac{p_\theta(x^l|c)}{p_{\text{ref}}(x^l|c)}\right)\right] \tag{2}$$

where: $p_\theta(x^w|c)$ is the likelihood of generating win output $x^w$ given input $c$ under the fine-tuned model, $p_{\text{ref}}(x^w|c)$ is the likelihood under the reference (pretrained) model. $\beta$ is a temperature parameter that controls the sharpness of preference optimization. $\sigma(\cdot)$ is the sigmoid function ensuring a proper probability score.

According to the derivation in reference (Wallace et al., 2024), the training objective of Diffusion-DPO is defined as:

$$L(\theta) = -\mathbb{E}_{(\boldsymbol{x}_0^w,\boldsymbol{x}_0^l)\sim D,t\sim\mathcal{U}(0,T),x_t^w\sim q(x_t^w|x_0^w),x_t^l\sim q(x_t^l|x_0^l)}\log\sigma\left(-\beta T\omega(\lambda_t)\right(
$$
$$\|\boldsymbol{\epsilon}^w - \boldsymbol{\epsilon}_\theta(x_t^w,t)\|_2^2 - \|\boldsymbol{\epsilon}^w - \boldsymbol{\epsilon}_{\text{ref}}(x_t^w,t)\|_2^2 - \left(\|\boldsymbol{\epsilon}^l - \boldsymbol{\epsilon}_\theta(x_t^l,t)\|_2^2 - \|\boldsymbol{\epsilon}^l - \boldsymbol{\epsilon}_{\text{ref}}(x_t^l,t)\|_2^2\right))) \tag{3}$$

where $\boldsymbol{x}_t^w = \alpha_t\boldsymbol{x}_0^w + \sigma_t\boldsymbol{\epsilon}^w, \boldsymbol{\epsilon}^w \sim \mathcal{N}(0,I)$ is a draw from $q\left(\boldsymbol{x}_t^w \mid \boldsymbol{x}_0^w\right)$. $\lambda_t = \alpha_t^2/\sigma_t^2$ is the signal-to-noise ratio. $\omega(\lambda_t)$ is a weighting function, usually kept constant.

## 4 THE REALDPO PARADIGM

In this section, we introduce our fine-tuning pipeline RealDPO to align video diffusion models with our constructed preferences data, as shown in Fig. 4. Firstly, we introduce our proposed dataset, RealAction, and the pipeline for constructing preference data in Sec.4.1. Then, in Sec.4.2, we present the win-lose sampling approach used for DPO fine-tuning training. Finally, in Sec.4.3, we delve into the alignment training process for the video generation model using preference data.

### 4.1 REALACTION: PREFERENCE DATA COLLECTION

Preference data is essential for reinforcement learning. To acquire it, we designed a robust data processing pipeline that efficiently collects, filters, and processes data, ensuring its high quality, diversity, and representativeness.

**Collect raw video based on keywords.** Our dataset construction begins with selecting relevant topics to collect raw video data, ensuring diversity and real-world representativeness. As shown in Fig. 3(d), we designed daily activity themes across over ten scenarios, such as sports, eating, drinking, walking, and gathered high-quality video clips using these keywords. This step captures diverse actions, participants, and backgrounds, establishing a strong foundation for preference-based training.

**Use VideoLLM to filter low-quality videos.** After collecting the raw videos, we use a video LLM, Qwen2-VL (Wang et al., 2024a) , to filter out rough or irrelevant videos. We provide some instructions for Qwen2-VL to identify and discard videos that do not meet quality standards or are not aligned with the selected topic. Through this filtering process, low-quality content is significantly reduced, ensuring that only clear and meaningful videos proceed to next processing stages.

**Manual inspection ensures the accuracy.** We let human annotators carefully examine the videos to confirm whether they accurately represent the intended theme, have correct actions, and do not contain misleading or irrelevant content. This additional validation step further refines the dataset, ensuring it aligns with preference-based training goals.

**Generate detailed descriptions for videos.** We employ a video understanding model, LLaVA-Video (Zhang et al., 2024b), to generate accurate descriptive captions for each video. These descriptions accurately reflect the actions, participants, and appearance. These captions serve as valuable metadata, later used for sampling negative samples. The word cloud composed of high-frequency words in the description caption of these videos is shown in Fig. 3(c). And the length distribution of captions in our constructed dataset is shown in the Fig 3(e).

## 4.2 WIN-LOSE SAMPLING FOR DPO TRAINING

After obtaining real data, we take the real video $X^w$ from RealAction as win sample. The latent after compression through the VAE encoder is $X_0^w$. We design a *Timestep Selector* that randomly generates a timestep k for each round of positive and negative sampling. We add k steps of random noise to $X_0^w$, obtaining $X_k^w$. Then, together with the caption embedding $e^{text}$, we input this into the DiT transformer to get the predicted noise $\hat{\epsilon}_k^w$. Finally, we input $\hat{\epsilon}_k^w$ to *Positive Sample Velocity* to obtain the predicted latent $\hat{x}_0^w$, which is prepared for the subsequent DPO loss.

$$\hat{\epsilon}_k^w = \theta\left(x_k^w, e^{text}\right), \hat{x}_0^w = \psi\left(\hat{\epsilon}_k^w\right), \tag{4}$$

where $\theta$ is the training DiT model, $\psi$ is the process of positive sample velocity.

For negative samples, in order to ensure diversity, we first randomly generate three init noises $\epsilon^a$, $\epsilon^b$, $\epsilon^c$. These are then combined with the positive sample's caption embedding $e^{text}$ and we input them together into the DiT, where we sample the full timesteps to obtain three negative samples $x_0^{l,a}$, $x_0^{l,b}$, $x_0^{l,c}$. This step is done offline, and we only need to store the latent of the negative samples. During training optimization, similar to the positive samples, we add k steps of random noise to these three samples to obtain $x_k^{l,a}$, $x_k^{l,b}$, $x_k^{l,c}$. Then, together with caption embedding $e^{text}$, we input these into the DiT transformer to get the predicted noise $\hat{\epsilon}_k^{l,a}, \hat{\epsilon}_k^{l,b}, \hat{\epsilon}_k^{l,c}$. Finally, we input predicted noise to *Negative Sample Velocity* to obtain the predicted latents for the negative samples $\hat{x}_0^{l,a}, \hat{x}_0^{l,b}, \hat{x}_0^{l,c}$.

$$\hat{\epsilon}_k^{l,*} = \theta\left(x_k^{l,*}, e^{text}\right), \hat{x}_0^{l,*} = \psi\left(\hat{\epsilon}_k^{l,*}\right), \tag{5}$$

where $*$ is set of $\{a, b, c\}$, $\psi$ is the process of negative sample velocity.

It's important to note that the first sampling of the negative samples is done offline, while the second sampling for both positive and negative samples involves only one step, saving a significant amount of time during training.

## 4.3 PREFERENCE LEARNING FOR VIDEO GENERATION

After positive and negative samples are prepared, we can use this preference data for DPO training. Due to the constraints of the reference model in DPO training, similarly, we also resample the win-lose samples through the reference model to obtain $\tilde{x}_0^w$ and $\tilde{x}_0^{l,a}$. Here, we take the first negative

Table 1: **Quantitative Comparison on RealAction-TestBench by User Study**. We provided users with a five dimensional evaluation, namely Overall Quality, Visual Alignment, Text Alignment, Motion Quality and Human Quality, to compare our model with the pre-trained baseline (Yang et al., 2024b), supervised fine-tuning(SFT), LiFT (Wang et al., 2024c), VideoAlign (Liu et al., 2025). Testers are required to rank the results generated by these models, and we converted the rankings into win rates.

| Method | Overall Quality | Visual Alignment | Text Alignment | Motion Quality | Human Quality |
|---|---|---|---|---|---|
| Baseline (Yang et al., 2024b) | 65.56 | 72.22 | 71.89 | 65.56 | 66.00 |
| SFT | 58.22 | 65.22 | 68.44 | 59.11 | 60.33 |
| LiFT (Wang et al., 2024c) | 67.34 | 73.44 | 64.33 | 65.00 | 67.33 |
| VideoAlign (Liu et al., 2025) | 61.00 | 68.11 | 68.78 | 57.22 | 59.78 |
| RealDPO (Ours) | **73.33** | **77.44** | **77.00** | **71.00** | **72.89** |

*"The video begins with a man wearing a white shirt and blue tie, sitting at an outdoor table. He holds a cup of coffee in one hand, taking a sip, while his other hand is occupied with writing notes on a notepad. The table also holds a smartphone and a pair of sunglasses, suggesting a busy day ahead. The setting is an urban street scene with brick walls and parked cars in the background, providing a realistic and bustling atmosphere. The scene captures a moment of focus and productivity amidst the daily hustle."*

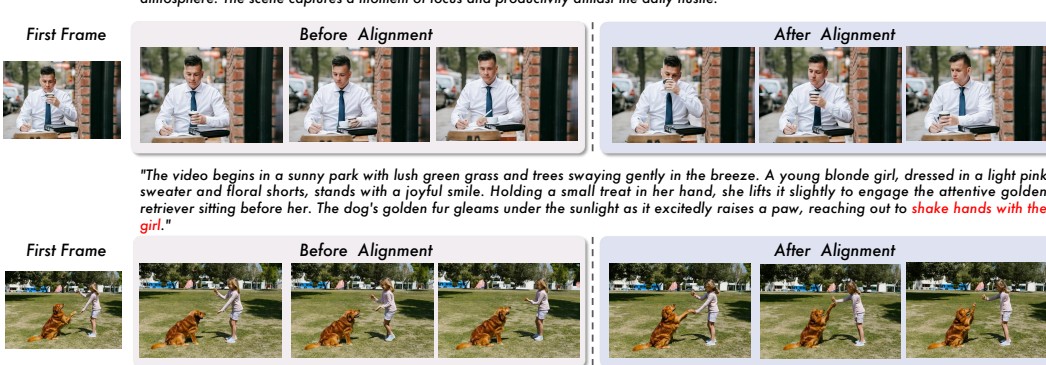

*"The video begins in a sunny park with lush green grass and trees swaying gently in the breeze. A young blonde girl, dressed in a light pink sweater and floral shorts, stands with a joyful smile. Holding a small treat in her hand, she lifts it slightly to engage the attentive golden retriever sitting before her. The dog's golden fur gleams under the sunlight as it excitedly raises a paw, reaching out to shake hands with the girl."*

Figure 5: **Qualitative Results.** We visualize the effect of before and after applying RealDPO. See the supplementary for videos.

sample $\tilde{x}_0^{l,a}$ as an example to explain the loss. According to the training objective of CogVideoX-5B (Yang et al., 2024b), we rewrite Eq. 3 as follows:

$$L_{DPO}(\theta) = -\mathbb{E}\left[\log \sigma \left(-\beta T \omega(\lambda_t) \left(\|\boldsymbol{x}_0^w - \hat{\boldsymbol{x}}_0^w\|_2^2 - \|\boldsymbol{x}_0^w - \tilde{\boldsymbol{x}}_0^w\|_2^2 \right.\right.\right.$$
$$\left.\left.\left. - \left(\|\boldsymbol{x}_0^l - \hat{\boldsymbol{x}}_0^l\|_2^2 - \|\boldsymbol{x}_0^l - \tilde{\boldsymbol{x}}_0^l\|_2^2\right)\right)\right)\right], \tag{6}$$

where $x_0^w / x_0^l$ are the original win/lose sample, $\hat{x}_0^w / \hat{x}_0^l$ are the predicted latents for the win/lose sample by the *training model*, $\tilde{x}_0^w / \tilde{x}_0^l$ are the predicted latents for the win/lose sample by the *reference model*. The role of the reference model is to constrain the training process of the training model, preventing over-optimization or deviation from the desired objectives.

To enhance alignment of the model with human preferences, we gradually improve the capability of the preference model and perform multiple rounds of resampling, ensuring that the training process iteratively refines its predictions and better captures the desired outcomes. In practice, every $t$ training steps, the reference model needs to be updated using the exponential moving average (EMA) algorithm.

$$\theta_{\mathrm{ref}}^t \leftarrow \omega \theta_{\mathrm{ref}}^t + (1 - \omega)\theta^t, \tag{7}$$

where $\theta_{\mathrm{ref}}^t$ is the parameters of the reference model, $\theta^t$ denotes the parameters of the training model, and $\omega$ is the decay coefficient of EMA, set to 0.996 in our experiments.

Table 2: **Quantitative Comparison on VBench-I2V and RealAction-TestBench using MLLM.** We evaluate performance via Visual Alignment (VA), Text Alignment (TA), Motion Quality (MQ), and Human Quality (HQ), which are consistent with the sensory perceptions of humans in user study. We use open-source Qwen2-VL (Wang et al., 2024a) supporting video understanding. We provide a detailed instruction template for evaluating video generation quality using MLLM in the appendix.

| Method | VBench-I2V | | | | RealAction-Test Bench | | | |
|---|---|---|---|---|---|---|---|---|
| | VA ↑ | TA ↑ | MQ ↑ | HQ ↑ | VA ↑ | TA ↑ | MQ ↑ | HQ ↑ |
| Baseline (Yang et al., 2024b) | 97.78% | 97.71% | 89.86% | 90.34% | 96.11% | **99.22%** | 90.22% | 91.89% |
| SFT | 97.15% | **98.26%** | **90.03%** | 89.38% | 93.89% | 98.89% | 90.78% | 92.89% |
| LiFT (Wang et al., 2024c) | 97.54% | 97.91% | 89.25% | 90.24% | **97.54%** | 97.89% | **92.00%** | 91.67% |
| VideoAlign (Liu et al., 2025) | **97.99%** | 97.66% | 89.54% | **90.84%** | 96.44% | 98.89% | **92.00%** | 92.89% |
| RealDPO (Ours) | **97.99%** | 97.74% | 89.46% | 90.10% | 96.67% | **99.22%** | 91.67% | **94.11%** |

Table 3: **Quantitative Comparisons** with baselines and reward-based methods via VBench-I2V (Huang et al., 2024a;b), on RealAction-TestBench.

| Model | I2V Subject | Subject Consistency | Background Consistency | Motion Smoothness | Dynamic Degree | Aesthetic Quality | Imaging Quality |
|---|---|---|---|---|---|---|---|
| Baseline (Yang et al., 2024b) | 96.10 | 90.43 | 94.01 | 98.15 | 55.56 | 59.63 | 67.01 |
| SFT | 96.47 | 89.50 | 93.18 | 98.06 | **66.67** | 59.69 | 67.06 |
| LiFT (Wang et al., 2024c) | 96.50 | **92.34** | 94.46 | 98.20 | 38.89 | 60.51 | **68.40** |
| VideoAlign (Liu et al., 2025) | 96.55 | 92.23 | 94.29 | **98.37** | 50.00 | 60.21 | 67.66 |
| RealDPO (Ours) | **96.58** | 91.68 | **94.47** | 98.31 | 55.56 | **61.37** | 68.05 |

(left margin, rotated: CogvideoX-5B)

## 5 EXPERIMENTS

We present the main experiments and discussions in this section. Please refer to the supplementary material for implementation details on the models and evaluation metrics.

### 5.1 QUANTITATIVE COMPARISONS

**Quantitative Comparison by User Study.** Tab. 1 showcases the evaluation results on the RealAction-TestBench test set, where testers were invited to rank the generated outputs of the pre-trained baseline (Yang et al., 2024b), supervised fine-tuning (SFT), LiFT (Wang et al., 2024c), VideoAlign (Liu et al., 2025), and our RealDPO. The evaluation covers five dimensions: Overall Quality, Visual Alignment, Text Alignment, Motion Quality, and Human Quality. The scores for each model across these dimensions were calculated and summarized. As shown in Tab. 1, our RealDPO demonstrates significant improvements over baseline and SFT in multiple dimensions, indicating that our proposed data effectively enhance the capabilities of RealDPO in action-centric scenarios. Additionally, compared to other preference alignment algorithms utilizing reward models, such as LiFT (based on Reward Weighted Regression) and VideoAlign (a naive DPO variant using synthetic data), our approach of leveraging real data as win samples and synthetic data as lose samples also proves its effectiveness.

**Quantitative Comparison Using MLLM.** To enhance the diversity of evaluation, we employ MLLM capable of video understanding tasks to assess the results generated by the models in a question-answer format across multiple dimensions. In Tab. 2, we selected Qwen2-VL (Wang et al., 2024a) as an evaluation tool, to align the assessment dimensions with user study: Visual Alignment (VA), Text Alignment (TA), Motion Quality (MQ), and Human Quality (HQ) on VBench-I2V test benchmark (Huang et al., 2024b) and RealAction-TestBench. For each dimension, we designed several questions, and a "yes" response from the large models indicates a passing score. The scores for all questions within each dimension were aggregated to calculate the total score. Experimental results show that, based on the evaluation by video-language understanding models, our model shows

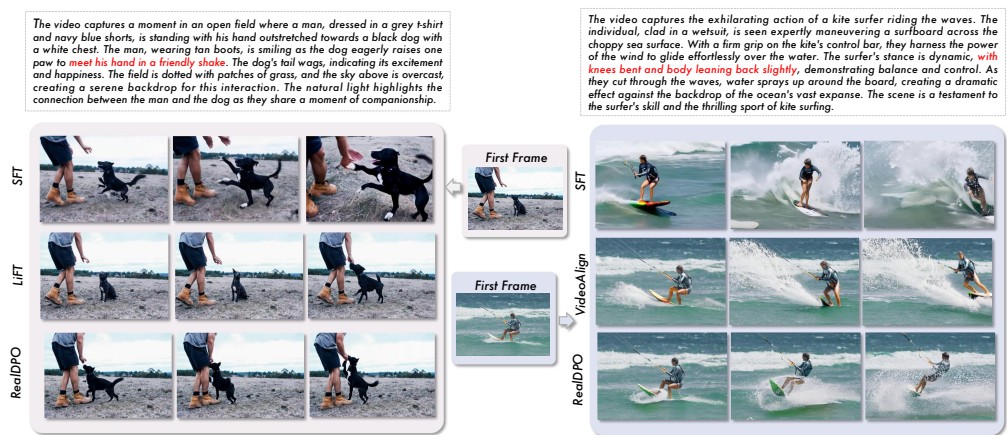

Figure 6: **Qualitative Comparison.** We recommend readers refer to our appendix files to view more visualizations.

competitive results, consistent with human evaluation results, further validating the effectiveness of our RealDPO.

**Quantitative Comparison Using VBench-I2V Metric.** Meanwhile, in video generation, VBench (Huang et al., 2024a;b) is widely recognized as an authoritative evaluation framework. Leveraging VBench-I2V's automated metrics designed for Image-to-Video (I2V) evaluations in VBench++ (Huang et al., 2024b), we assessed the quality of our test set, revealing that RealDPO achieves competitive performance across multiple general metrics.

## 5.2 QUALITATIVE COMPARISONS

In Fig. 5, we present the visual comparison results before and after RealDPO alignment. We observe that RealDPO is highly effective in enhancing the naturalness and smoothness of the actions, as well as their consistency with the textual instructions. In Fig. 6, we present the visual comparison results of our method against other alignment approaches, such as LiFT (Wang et al., 2024c) and VideoAlign (Liu et al., 2025). It can be observed that the videos generated by RealDPO are more stable and less prone to unnatural actions or visual collapse. For instance, in the example on the left, SFT exhibits a collapse of the character's limbs, and the coordination of the dog's four legs appears unnatural. The results of LiFT are slightly better, but LiFT fails to complete the handshake action between the protagonist and the dog, resulting in poor alignment with the text. In contrast, our results demonstrate higher visual quality, with action details highly consistent with the textual instructions and no visual collapse. In the example on the right, the text describes the surfer's posture as "with knees bent and body leaning back slightly". SFT shows visual collapse, misaligned actions, and poor consistency in character appearance. VideoAlign performs slightly better, but the generated posture and actions are not highly aligned with the text. In comparison, our results exhibit higher image quality, more accurate action details, and overall superior performance.

## 6 CONCLUSION

In this paper, we propose RealDPO, a novel and data-efficient framework for preference alignment in video generation, leveraging real-world data as win samples to address challenges in generating complex motions like human actions. By designing a tailored DPO loss and building on diffusion-based transformer architectures, we establish a robust real-data-driven alignment framework. To support this, we introduce RealAction-5K, a compact yet high-quality dataset for human daily actions. Extensive experiments show that RealDPO significantly improves visual alignment, text alignment, motion quality, and overall video quality, outperforming traditional fine-tuning and other alignment methods. Our work advances the upper bound of preference alignment and provides a scalable solution for complex motion video generation. We will explore extending RealDPO to broader domains in the future.

**Ethics statement.** Our RealAction-5K dataset was curated from publicly available video sources with appropriate licenses. To address privacy concerns, all personally identifiable information was meticulously anonymized, and the dataset will be released strictly for non-commercial research purposes to mitigate the risk of misuse. All necessary legal and ethical guidelines concerning data provenance and usage were adhered to throughout the project. Additionally, the effectiveness of our RealDPO paradigm is inherently limited by the architectural constraints of the underlying video generative model. We emphasize the need for responsible use, particularly when generating human figures, to prevent potential misuse.

**Reproducibility statement.** To ensure the reproducibility of our work, we have made significant efforts to document our methodology and resources comprehensively. The core of our approach, the RealDPO alignment paradigm, including its tailored loss function, is described in detail within the paper. Furthermore, we provide a complete account of the data collection and processing pipeline for the RealAction-5K dataset. This dataset was meticulously curated by manually sourcing high-quality videos from `https://pexels.com`. The process involved a combination of targeted scraping and manual downloading, followed by rigorous manual screening and clipping to ensure each video clip depicts a single, coherent action that can be accurately described in text, thereby guaranteeing high quality and clarity.

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

APPENDIX

This supplementary material provides more qualitative results, details of the evaluation, experimental results, pseudo-code of RealDPO. Section A elaborates on additional visual comparisons of generated videos, including comparisons with pre-trained models, supervised fine-tuning, and other alignment methods. Section B details the evaluation process, covering the design of user studies, evaluation using LLMs, evaluation using the VBench-I2V metric, as well as interfaces, instructions, and an introduction to automated evaluation metrics. Section C presents the pseudo-code of our core algorithm, RealDPO.

## A MORE QUALITATIVE RESULTS

Due to space limitations in the main text, this section presents additional visual comparisons, including comparisons with pre-trained models, supervised fine-tuning, and other alignment methods. The results demonstrate that our approach achieves superior performance across a wider range of samples, with enhanced visual-text alignment, text alignment, motion quality, character quality, and overall quality. These findings further validate the effectiveness of the RealDPO framework. This provides new insights and methodologies for future multi-modal generation tasks.

### A.1 COMPARISON WITH PRE-TRAINED MODEL

Pre-trained base models are typically trained on large-scale datasets and exhibit strong generalization capabilities. However, they may underperform on specific tasks, particularly those requiring fine-grained alignment, such as the generation of videos with complex motion as discussed in our work. RealDPO, by incorporating guidance from real-world data through Direct Preference Optimization (DPO), excels at capturing intricate details in tasks, especially in image-text alignment. As shown in Fig. 7, compared to pre-trained models, RealDPO demonstrates significantly improved consistency in visual-text alignment and notable enhancements in the details of characters and motions.

### A.2 COMPARISON WITH SUPERVISED FINE-TUNING

Supervised fine-tuning relies on annotated data and can achieve strong performance on specific tasks. However, its effectiveness is constrained by the quality and quantity of the available annotations. In contrast, RealDPO leverages a diverse set of negative samples and real-world data as positive samples to form multiple preference pairs. This approach enables the model to learn from its own mistakes and align more closely with real-world samples, achieving robust alignment even without extensive labeled data. In particular, as shown in Fig. 8, in terms of motion quality and character quality, RealDPO generates images that are more natural, with smoother motions and richer character details.

### A.3 COMPARISON WITH OTHER ALIGNMENT METHOD

Other reward model based alignment methods, such as LiFT and VideoAlign, may perform well on specific tasks. However, in complex scenarios, the reward models often fail to provide effective feedback, leading to misguidance in preference alignment training. In contrast, RealDPO introduces real-world samples as positive examples and pairs them with multiple negative samples generated by the model, naturally forming contrastive pairs. By guiding the model with positive samples, RealDPO enables the model to learn from its mistakes and align more closely with the correct samples, thereby better handling complex cross-modal alignment tasks. As shown in Figure 9, compared to existing alignment methods, RealDPO demonstrates greater stability in visual-text alignment and text alignment, generating images and text that are more semantically consistent, with superior motion quality.

*"The video begins with a person in grey shorts standing in a lush backyard, their hand outstretched towards a playful spotted puppy. The puppy, with its ears perked up and tongue out, eagerly raises one paw to meet the person's hand in a friendly gesture. The puppy's eyes are focused on the person's hand, and its tail wags with excitement. The backyard is filled with vibrant green grass and a wooden fence, creating a warm and inviting atmosphere for this interaction. The natural light from the setting sun casts a golden hue over the scene, highlighting the connection between the person and the puppy as they share a moment of joy and companionship."*

*"The video features a blonde skateboarder, dressed in a black jacket and shorts, standing in the middle of a road that stretches into the distance. They have one foot on the skateboard, ready to push off and begin skating. The road is bordered by snow-covered hills under a twilight sky, giving a sense of solitude and adventure. The anticipation of the skateboarder's impending journey is palpable."*

Figure 7: **Qualitative Results.** Comparison with pre-trained model.

*"The video captures a woman with her hair tied back, wearing a sleeveless top and jeans, standing in a bedroom. She is in the process of making a bed, holding up a white pillow with both hands, seemingly about to place it on the bed. The bedroom has a calm and tidy appearance, with a large window allowing natural light to brighten the room. The focus is on the woman's task, highlighting the routine activity of homemaking."*

*"The video captured a couple enjoying a moment on the beach. The man is wearing a gray T-shirt and shorts, and the woman is wearing a white striped sun skirt, they are embracing When they embrace on the damp beach, gentle waves beat against their feet, creating a playful and romantic atmosphere. The cloudy sky and distant hills added a dramatic backdrop to their leisurely stroll, highlighting the connection between the two and the beauty of the natural environment."*

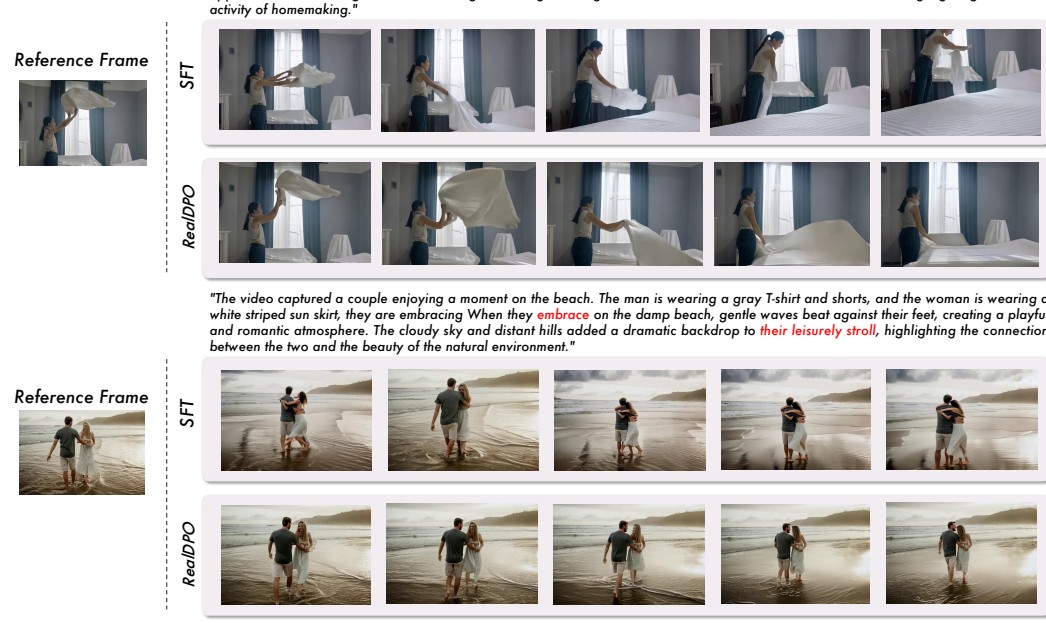

Figure 8: **Qualitative Results.** Comparison with supervised fine-tuning.

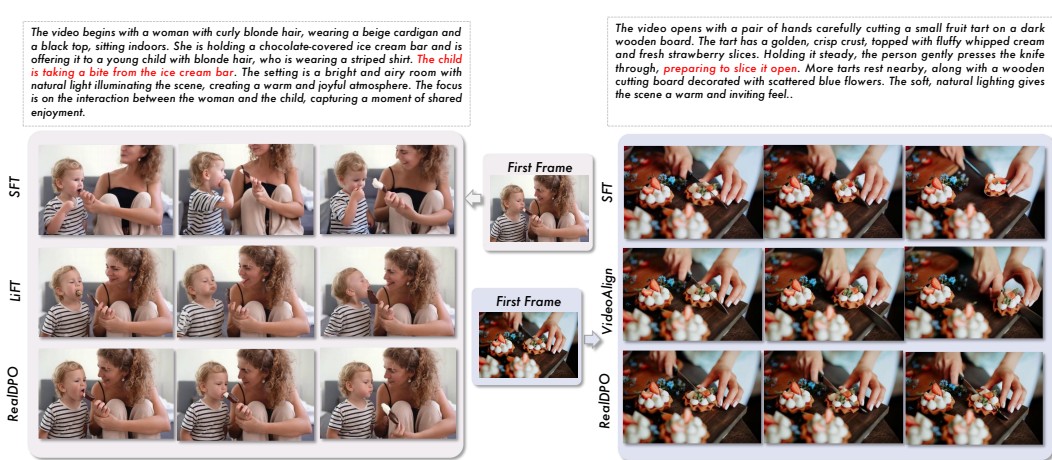

Figure 9: **Qualitative Results.** comparison with other Alignment Method.

## B DETAILS OF THE EVALUATION

### B.1 IMPLEMENTATION DETAILS

**Models and Settings.** We conduct all experiments on 8 Nvidia H100 GPUs, with a total batch size of 8 for training. For our I2V baseline generation model, we adopt CogVideoX-5B (Yang et al., 2024b), which uses diffusion transformer structure. We fine-tune the parameters of all its transformer blocks on the DeepSpeed framework. The learning rate is set to 1e-5, and all the models are trained for 10 epochs.

**Evaluation Metric.** We evaluate the performance of our aligned model through three aspects: user study, automatic LLM-based evaluation, and the assessment metrics of VBench (Huang et al., 2024a). We selected 18 test cases, including test texts and reference images, which constitute the RealAction-TestBench. For the user study, we invited 10 testers to evaluate our model against other baselines across multiple dimensions. For LLM-based evaluation, we designed a question template to guide the model in making decisions. As for VBench, we utilized the I2V evaluation metrics provided by VBench to perform our evaluation.

### B.2 EVALUATION BY USER STUDY

To ensure a fair and comprehensive evaluation of our model, we conducted a user study to collect subjective feedback on the generated results. Participants were presented with a scoring interface (as shown in Fig. 10) and asked to rate the generated videos based on several key criteria, including Visual Alignment, Text Alignment, Motion Quality and Human Quality and Overall Quality. The interface is designed to be intuitive and user-friendly, allowing participants to provide accurate and unbiased scores. Each video was evaluated by multiple users, and the final scores were averaged to ensure reliability.

### B.3 EVALUATION USING LLMS

In addition to human evaluation, we leveraged large language models (LLMs) to assess the quality of the generated videos. We designed a structured instruction template to guide the LLMs in evaluating video generation quality. The template includes detailed prompts for assessing various aspects of the videos, such as adherence to textual descriptions, visual coherence, and overall aesthetic appeal. By utilizing LLMs, we were able to obtain scalable and consistent evaluations that complement the human user study. The results from the LLM-based evaluation align closely with the user study findings, further validating the effectiveness of our approach

## B.4 EVALUATION USING VBENCH-I2V METRIC

To provide a more objective and fine-grained assessment of our model's performance, we select seven theme-related and human-perception-aligned representative dimensions of video quality from VBench-I2V Huang et al. (2024a;b) as the final evaluation metrics: *I2V Subject, Subject Consistency, Background Consistency, Motion Smoothness, Dynamic Degree, Aesthetic Quality, and Imaging Quality*.

Figure 10: User Study Scoring Interface for users to give socre.

**Instruction Template for Evaluating Video Generation Quality Using LLMs (part1)**

As a video understanding expert, you will be required to evaluate the quality of model-generated videos from four different perspectives, covering the following daily human activities. The specific evaluation angles will be divided into Visual Alignment, Text Alignment, Motion Quality, and Human Quality. Under each dimension, the model needs to determine the quality of the generated content based on the answers to five questions. Each question is scored out of 10 points. The scoring rule is 0 points for the worst and 10 points for the best. The final score is the sum of the scores for all questions under that dimension.

**Visual Alignment**
This mainly assesses the consistency of the visual representation of the characters in the generated video with the provided first frame image of the characters and environment, with a score range of 0 to 10. Please answer five questions as follow:
Question 1: What is the consistency score of the character's appearance (such as clothing, hairstyle, skin color) in the generated video?
Question 2: What is the consistency score of the environment in the generated video (such as background, lighting, scene setup)? Question 3: What is the score for the character's proportional changes in the generated video conforming to physical laws?
Question 4: What is the fidelity score of the characters or environment in the video (low scores should be given if there are shape distortions or color abnormalities)?
Question 5: What is the consistency score of the characters and environment over time (low scores should be given if there are sudden disappearances or changes)?

Answer 1:
Answer 2:
Answer 3:
Answer 4:
Answer 5:

**Instruction Template for Evaluating Video Generation Quality Using LLMs (part2)**

As a video understanding expert, you will be required to evaluate the quality of model-generated videos from four different perspectives, covering the following daily human activities. The specific evaluation angles will be divided into Visual Alignment, Text Alignment, Motion Quality, and Human Quality. Under each dimension, the model needs to determine the quality of the generated content based on the answers to five questions. Each question is scored out of 10 points. The scoring rule is 0 points for the worst and 10 points for the best. The final score is the sum of the scores for all questions under that dimension.

**Text Alignment**
This assesses the consistency between the actions of the characters in the generated video and the input text description or target behavior category, with a score range of 0 to 10. Please answer five questions as follow:
Question 1: What is the consistency score between the character's actions in the generated video and the input text description or target behavior category?
Question 2: What is the consistency score between the key actions in the video (such as running, hugging, playing an instrument) and the text description?
Question 3: What is the score for avoiding actions or distracting elements in the video that are unrelated to the text description?
Question 4: What is the accuracy score of the video in conveying the emotions or intentions described in the text?
Question 5: What is the score for the video supplementing reasonable details not explicitly mentioned in the text description?

Answer 1:
Answer 2:
Answer 3:
Answer 4:
Answer 5:

972
973
974
975
976
977
978
979
980
981
982
983
984
985
986
987
988
989
990
991
992
993
994
995
996
997
998
999
1000
1001
1002
1003
1004
1005
1006
1007
1008
1009
1010
1011
1012
1013
1014
1015
1016
1017
1018
1019
1020
1021
1022
1023
1024
1025

**Instruction Template for Evaluating Video Generation Quality Using LLMs (part3)**

As a video understanding expert, you will be required to evaluate the quality of model-generated videos from four different perspectives, covering the following daily human activities. The specific evaluation angles will be divided into Visual Alignment, Text Alignment, Motion Quality, and Human Quality. Under each dimension, the model needs to determine the quality of the generated content based on the answers to five questions. Each question is scored out of 10 points. The scoring rule is 0 points for the worst and 10 points for the best. The final score is the sum of the scores for all questions under that dimension.

**Motion Quality**
This assesses the smoothness, naturalness, and reasonableness of the character's movements in the generated video, with a score range of 0 to 10. Please answer five questions as follow:
Question 1: What is the smoothness score of the character's movements in the generated video (high scores for no stuttering)?
Question 2: What is the score for the details of the character's movements (such as limb movements, gestures) conforming to physical laws?
Question 3: What is the naturalness score of the temporal dynamics of the character's movements (such as speed, rhythm)?
Question 4: What is the coordination score between the character's movements and other elements in the scene (such as objects, background)?
Question 5: What is the score for avoiding obvious distortions or unreasonable phenomena in the character's movements (such as limb twisting, incoherent actions)?

Answer 1:
Answer 2:
Answer 3:
Answer 4:
Answer 5:

1026
1027
1028
1029
1030
1031
1032
1033
1034
1035
1036
1037
1038
1039
1040
1041
1042
1043
1044
1045
1046
1047
1048
1049
1050
1051
1052
1053
1054
1055
1056
1057
1058
1059
1060
1061
1062
1063
1064
1065
1066
1067
1068
1069
1070
1071
1072
1073
1074
1075
1076
1077
1078
1079

---

**Instruction Template for Evaluating Video Generation Quality Using LLMs (part4)**

As a video understanding expert, you will be required to evaluate the quality of model-generated videos from four different perspectives, covering the following daily human activities. The specific evaluation angles will be divided into Visual Alignment, Text Alignment, Motion Quality, and Human Quality. Under each dimension, the model needs to determine the quality of the generated content based on the answers to five questions. Each question is scored out of 10 points. The scoring rule is 0 points for the worst and 10 points for the best. The final score is the sum of the scores for all questions under that dimension.

**Human Quality**
This assesses the quality of the generated characters in the video, with a score range of 0 to 10.
Please answer five questions as follow:
Question 1: What is the score for the reasonableness of limb distortions in the generated characters (e.g., unnatural joint bends)?
Question 2: What is the score for the reasonableness of the number of limbs in the characters (e.g., extra or missing limbs)?
Question 3: What is the naturalness score of the facial expressions or body movements of the characters in line with human behavioral characteristics?
Question 4: What is the reasonableness score of the interactions between the characters and other objects in the scene (e.g., tools, animals, other people)?
Question 5: What is the fluency score of the characters' behavior in the generated video?

Answer 1:
Answer 2:
Answer 3:
Answer 4:
Answer 5:

## C PSEUDO-CODE OF REALDPO

```python
def RealDPO_Loss(model, ref_model, x_w, x_l, c, beta):
    """
    Computes the RealDPO loss for aligning model predictions with
        preferred and non-preferred samples.

    Args:
        model: Diffusion Transformer model.
        ref_model: Frozen reference model used for comparison.
        x_w: Preferred real video latents (aligned with the desired
            output).
        x_l: Non-preferred video-generated video latents (not aligned
            with the desired output).
        c: Conditioning input (e.g., text embeddings, image embeddings).
        beta: Regularization parameter controlling the strength of the
            alignment.

    Returns:
        realdpo_loss: The computed RealDPO loss value.
    """
    # Sample random timesteps and noise for diffusion process
    timestep_k = torch.rand(len(x_w))
    noise = torch.randn_like(x_w)

    # Create noisy versions of preferred and non-preferred latents
    noisy_x_w = (1 - timestep_k) * x_w + timestep_k * noise
    noisy_x_l = (1 - timestep_k) * x_l + timestep_k * noise

    # Predict latents using the model and reference model
    latent_w_pred = model(noisy_x_w, c, timestep_k)
    latent_l_pred = model(noisy_x_l, c, timestep_k)
    latent_ref_w_pred = ref_model(noisy_x_w, c, timestep_k)
    latent_ref_l_pred = ref_model(noisy_x_l, c, timestep_k)

    # Compute prediction errors for preferred and non-preferred latents
    model_w_loss = (x_w - latent_w_pred).norm().pow(2)
    ref_w_loss = (x_w - latent_ref_w_pred).norm().pow(2)
    model_l_loss = (x_l - latent_l_pred).norm().pow(2)
    ref_l_loss = (x_l - latent_ref_l_pred).norm().pow(2)

    # Compute alignment differences
    w_loss_diff = model_w_loss - ref_w_loss
    l_loss_diff = model_l_loss - ref_l_loss

    # Compute the RealDPO loss
    alignment_term = -0.5 * beta * (w_loss_diff - l_loss_diff)
    realdpo_loss = -1 * torch.log(torch.sigmoid(alignment_term))

    return realdpo_loss
```

# D   THE USE OF LARGE LANGUAGE MODELS (LLMS)

In the preparation of this manuscript, we used GPT-4, a large language model from OpenAI, exclusively as a writing assistance tool. Its use was confined to the Introduction and Methods sections, where it served to aid in polishing the text. Specifically, the model was prompted to help restructure sentences for improved clarity and flow, ensure consistent academic tone, and simplify complex technical descriptions. All fundamental ideas, research hypotheses, methodological designs, experimental data, analysis, conclusions, and the final intellectual content are solely the product of the authors' work. The LLM generated no original content or ideas and was not used for data analysis or interpretation. The authors carefully reviewed, edited, and verified all AI-generated text to ensure it accurately reflected their research and adhered to the highest standards of academic integrity.

