# OpenReview forum: "RealDPO: Real or Not Real, that is the Preference"
_ICLR.cc/2026/Conference — ICLR 2026 Conference Withdrawn Submission_

### Official Review · Reviewer_chqu · 2025-10-27

**Soundness:** 3
**Presentation:** 3
**Contribution:** 3
**Rating:** 4
**Confidence:** 2

**Summary:**

The authors introduce RealDPO for improving the realism and quality of generated videos focusing on complex motion. Methods like supervised fine tuning and RLHF suffer from reward hacking, poor comparison signals, and distribution gaps. The authors propose a DPO approach in which the use of real, high-quality videos work as the win samples and the model's own generated videos are lose samples. This guides the model to differentiate its outputs from realistic motion. The authors also introduce a new dataset, RealAction-5K containing ~5,000 human action videos with manual verification.The experiments show that RealDPO outperforms SFT and other preference alignment baselines.

**Strengths:**

1.  The central idea of using real videos as the win samples is novel and addresses a fundamental weakness of existing preference learning methods for generation.
2.  The authors clearly identify the critical challenge of generating realistic, complex motion in videos and convincingly argue why current methods fall short. RealDPO offers an intuitive solution to this problem.
3. The RealAction-5K dataset is a good contribution with focus on high-quality, action-centric clips.
4. The experiments show that the proposed method outperforms the considered baselines.

**Weaknesses:**

1. The lose samples may always produce flaws of a certain kind which can get corrected over time, but some other flaws which were never generated during training, which could be due to limited training data, may never get corrected. More discussion on the diversity and potential bias of generated lose samples should make the work stronger.
2. The win samples have specific motions. For a prompt that is very different from anything in RealAction-5K, will the model still generalize the learned motion realism?
3. In Tables 3 and 4, the proposed method does not provide much gain over the motion quality compared to the baseline methods. Also, there is no clear gain across the remaining metrics as well.

**Questions:**

1. Why was Qwen2-VL used, there are better Open-source models like Qwen2.5-VL, Intern3-VL?
2. For human evaluation, how many videos were evaluated? I am not able to understand Table 1. What do those numbers mean? From the appendix, it appears that each video is scored on a scale of 1-5, so how are the scores calculated from this?

**Details Of Ethics Concerns:**

The authors collect a dataset of 5k videos by crawling the web. I am not sure of the licensing of the original videos.

---

### Official Review · Reviewer_RBi3 · 2025-10-29

**Soundness:** 2
**Presentation:** 2
**Contribution:** 2
**Rating:** 4
**Confidence:** 3

**Summary:**

In this paper, it utilized real video as preference for video diffusion model fine-tuning. In addition, it collected over 5k video to propose a dataset, termed as RealAction-5K. The proposed method is validated on a existing dataset and the proposed dataset to show the performance.

**Strengths:**

1. It has collected more annotated videos to form a new dataset.
2. The proposed method presented better performance than some previous works.

**Weaknesses:**

1. The size of the proposed dataset is limited, only suitable for fine-tuning.
2. According to the data processing pipeline, no human filtering is taken to ensure all annotations are correct.
3. Utilizing real data as win sample is not novel to me.

**Questions:**

1. Whether the proposed method contributed more to the performance improvements. The performance boost over baseline is not obvious.

---

### Official Review · Reviewer_cXEF · 2025-10-30

**Soundness:** 2
**Presentation:** 2
**Contribution:** 2
**Rating:** 2
**Confidence:** 4

**Summary:**

The paper introduces RealDPO, a method for improving video generation by using real data as positive samples for preference learning, enhancing motion realism through a tailored DPO loss and outperforming existing models in quality and alignment.

**Strengths:**

1.  the paper is well structured and visual appealing.
2.  It provides a real-world dataset that could benefit the community if open-sourced.
3.  The construction pipeline for the dataset is clearly introduced.

**Weaknesses:**

Main concerns:
1. High-Risk Assumption in Negative Sampling: The most critical issue is the use of all generated videos as negative samples, which is highly risky, especially for DPO-based training strategies. RealDPO assumes that the current model's generation domain is entirely misaligned with the requirements and forcibly binds the generation domain to a fixed set of 5000 training domains. And this paper does not provide a detailed explanation or thorough ablation studies to justify why all model-generated results are considered poor. This assumption may not hold in practice and could limit the model's ability to learn effectively.

2. Limited Model Validation: The validation of the proposed method is somewhat limited, as the authors only use CogVideoX-5B as the backbone model. To strengthen the credibility of the results, it would be more convincing to conduct experiments on more advanced models such as HunyuanVideo and Wan2.1. This would help demonstrate the generalizability and robustness of the proposed method across different model architectures.

Data Construction:
1. Potential Bias in Data Filtering: In line 100, the authors mention issues related to reward hacking and bias. However, Qwen2-VL and LLaVA-Video are used for data filtering and caption generation. So how to ensure these models do not introduce bias into the data? Given that Qwen2-VL and LLaVA-Video are relatively outdated, their filtering and labeling quality may not be ideal. It would be more appropriate to use models like Qwen2.5VL or Genmi 2.5 pro for these tasks.

2. Narrow Domain of Real Data: The domain of the 5000 real data samples across 10 scenarios is relatively narrow. This may limit the model's generalization ability, especially when all generated samples are treated as negative samples. A broader and more diverse dataset would likely be more effective in training a model that can generalize well to unseen scenarios.

Experiments:
1. Questionable User Study: The user study involves only 10 testers evaluating 18 sample videos, which raises concerns about the reliability of the results. Additionally, the paper does not clearly explain how the rankings in Fig. 10 translate into the scores presented in Tab. 1. A more robust user study design with a larger number of participants and clearer methodology would enhance the credibility of the findings.
2. Lack of Detailed Description for RealAction-TestBench: The paper lacks a detailed description of the RealAction-TestBench dataset, which is crucial for understanding the evaluation framework and the results. More information about the dataset's composition, diversity, and relevance to real-world scenarios would be beneficial.
3. Insignificant Improvements in Quantitative Comparisons: In the quantitative comparisons presented in Tab. 2 and Tab. 3, the improvements achieved by RealDPO are not substantial and its overall impact on video generation quality may be limited compared to other methods.

Additional Suggestions:

1. The authors could include more discussions related to GRPO and comparisons with other Video-DPO methods. This would provide a more comprehensive view of how RealDPO fits into the broader landscape of video generation techniques.
2. The authors used the same Qwen2-VL model for both data construction and LLM-based evaluation. It would be better to use a different model for evaluation to avoid an unfair comparison.

**Questions:**

Please refer to the weakness.

---

### Official Review · Reviewer_1Vpr · 2025-10-31

**Soundness:** 2
**Presentation:** 2
**Contribution:** 2
**Rating:** 6
**Confidence:** 4

**Summary:**

Paper. RealDPO: Real or Not Real, That Is the Preference proposes a post‑training paradigm for action‑centric video generation that replaces reward models and synthetic pair mining with Direct Preference Optimization (DPO) where real videos are always the “win” and model‑generated videos are the “lose.” The method targets diffusion–transformer (DiT) video generators (e.g., CogVideoX‑5B) and introduces: (i) a tailored DPO loss in latent space aligned with the DiT training objective; (ii) an offline/online win–lose sampling pipeline that re‑noises positives and stored negatives at a random timestep k; (iii) an EMA‑updated reference model to prevent over‑optimization; and (iv) RealAction‑5K, a curated set of everyday human‑action videos with detailed captions used as positive data. Experiments include a 18‑prompt RealAction‑TestBench, user studies, MLLM‑based evaluation with Qwen2‑VL, and VBench‑I2V metrics. The paper reports improved motion realism and alignment versus supervised fine‑tuning (SFT) and reward‑based preference methods (LiFT, VideoAlign). See the overview and comparison in Fig. 1 (p. 1), framework in Fig. 4 (p. 5), and quantitative tables (Tab. 1–3, pp. 7–8).

**Strengths:**

• Originality: Clean, compelling idea—use real videos as the positive signal—to avoid reward hacking and ambiguous synthetic pair mining. The focus on human action quality targets a real pain point for current models.
• Quality: The training recipe is practical (latent-space, offline negatives, EMA reference) and easy to implement. The dataset and sampling details are clear enough to reproduce. The qualitative comparisons convincingly highlight fewer limb collapses and better action execution.
• Clarity: The paper is well organized, with an intuitive framework description and reader-friendly figures and pseudocode.
• Significance: If broadly validated, the “real-as-win” recipe could become a standard alignment tool for motion-centric video generation, offering a simpler alternative to reward-model pipelines.

**Weaknesses:**

• Evaluation scale and rigor: The main user study uses 18 prompts and 10 raters; scores are reported as win rates without confidence intervals, variance, or statistical tests. This limits the strength of the claims.
• Baselines: While SFT and a few reward-based methods are compared, other recent reward-free or video-DPO approaches are not included, making it hard to judge generality.
• Metrics: Automated VBench improvements are modest or mixed, which creates a gap between subjective wins and standard metrics; the paper does not analyze this discrepancy.
• Technical clarity: Some implementation details are ambiguous (e.g., whether positives and negatives share noise in sampling; exact schedules and constants for key hyperparameters).
• Ablations: Missing sensitivity studies on number of negatives per positive, EMA decay/update cadence, temperature, timestep selection, or dataset size.
• Scope: Results focus on image-to-video with a first frame and on human daily actions; generalization to text-to-video, non-human motion, or longer horizons remains untested.
• Data & ethics: Dataset licensing and privacy/consent specifics are brief; a fuller data card would improve transparency.
• Efficiency claims: The method is positioned as efficient, but wall-clock, throughput, or GPU-hour comparisons vs. reward-model pipelines are not reported.

**Questions:**

Please clarify the exact training choices: the temperature and weighting terms, whether they are fixed or scheduled, and how sensitive results are to them.

In the win/lose sampling, do positives and negatives use independent noise when re-noised at the selected step, or the same noise tensor? If shared, why, and what is the observed impact vs. independent noise?

Provide ablations: number of negatives per positive; EMA decay and update cadence; timestep selection strategy; and performance vs. the size of RealAction-5K.

Expand evaluation rigor: report confidence intervals, statistical tests, and the precise procedure for converting rater rankings to win rates. Consider enlarging the testbench and reporting category-wise results and held-out action classes.

Analyze the metric gap: why do human/MLLM preferences improve more than VBench submetrics? Which components fail to capture motion nuances, and what motion-specific metrics could better reflect the improvements?

Add stronger baselines: include recent reward-free preference methods for video under matched compute/data to support broader claims.

Test scope/transfer: show qualitative or quantitative evidence for text-to-video without a reference frame, non-human motion, and longer sequences.

Efficiency: report training time, throughput, and GPU hours compared with SFT and a typical reward-model alignment pipeline to substantiate the efficiency narrative.

Dataset card: spell out license terms, PII handling, age filtering, provenance, geographic diversity, opt-out procedures, and whether you will release original links/IDs for reproducibility.

Failure cases and safety: include a short section with typical failure modes (fast motion, occlusion, multi-person interactions) and discuss practical mitigations.

Overall recommendation
Lean accept. The idea is simple and timely, the engineering is sound, and the qualitative and user/MLLM results are promising for a real problem in video generation. To strengthen the paper, the authors should bolster evaluation rigor, broaden baselines, clarify implementation details, and provide a fuller dataset card and efficiency accounting

---

### Note · Authors · 2025-12-02

I have read and agree with the venue's withdrawal policy on behalf of myself and my co-authors.